# Which Attributes of Credibility Matter for Quality Improvement Projects in Hospital Care—A Multiple Case Study among Hospitalists in Training

**DOI:** 10.3390/ijerph192316335

**Published:** 2022-12-06

**Authors:** Lisanne Hut-Mossel, Kees Ahaus, Gera Welker, Rijk Gans

**Affiliations:** 1Department of Internal Medicine, University Medical Centre Groningen, University of Groningen, 9700 RB Groningen, The Netherlands; 2Department Health Services Management & Organisation, Erasmus School of Health Policy & Management, Erasmus University, 3062 PA Rotterdam, The Netherlands; 3UMC Staff Policy and Management Support, University Medical Centre Groningen, University of Groningen, 9700 RB Groningen, The Netherlands

**Keywords:** quality improvement, champions, hospitalist, training, education, clinical lead, leadership, implementation, clinical audit, hospital care

## Abstract

Healthcare professionals have to give substance to the role of a champion in order to successfully lead quality improvement (QI) initiatives. This study aims to unravel how hospitalists in training shape their role as a champion within the context of QI projects in hospital care and why some are more effective in leading a QI project than others. We focus on the role of credibility, as it is a prerequisite for fulfilling the role of champion. This multiple-case study builds upon 23 semi-structured interviews with hospitalists in training: quality officers and medical specialists. We first coded data for each case and then described the different contexts of each case in detail to enable comparison across settings. We then compared the cases and contrasted the attributes of credibility. Four attributes of credibility emerged and were identified as essential for the hospitalist in training to succeed as a champion: (1) being convincing about the need for change by providing supportive clinical evidence, (2) displaying competence in their clinical work and commitment to their tasks, (3) generating shared ownership of the QI project with other healthcare professionals, and (4) acting as a team player to foster collaboration during the QI project. We also identified two contextual factors that supported the credibility of the hospitalist in training: (1) choosing a subject for the QI project that was perceived as urgently required by the group of stakeholders involved, and (2) being supported by the board of directors and other formal and informal leaders as the leader of a QI project. Further research is needed to gain a deeper understanding of the relationship between credibility and sustainability of change.

## 1. Introduction

Healthcare professionals who successfully lead quality improvement (QI) initiatives are regarded as champions [1,2]. Across hospital and outpatient settings, such champions have facilitated QI initiatives by building support for change within their organization and among their colleagues [3,4,5,6,7,8]. Personal commitment, credibility, QI behaviours and skills, and institutional knowledge, through which champions instinctively navigate the culture of their organisation to overcome stakeholder resistance, have been identified as key characteristics of champions [3,4,6,9,10,11]. 

Previous research suggests that credibility might be the decisive factor in successfully fulfilling a champion’s role, and that champions earn credibility by demonstrating clinical knowledge that fosters trust among colleagues that the QI project is worth the effort [1,3,9,12]. From our conceptualization, credibility can be built over time and calls for the quality of being believable in the eyes of others, which is important in developing trust. Being credible as a champion to one’s colleagues is an important factor in stimulating, educating and leading them in QI projects, in being listened to and ‘in getting things done’ [1,12,13]. In addition to earning credibility as a champion, credibility needs to be granted by other healthcare professionals. Gaining a better understanding of how champions shape their role, how they earn and are granted credibility, as well as how and why champions are effective and in which organisational contexts, can contribute to gaining deeper insights into the specific approaches through which champions enact change within QI projects. The existing literature focuses on the strategies and resources of effective champions, but relatively little is known about the attributes of credibility that might be important for champions to be successful in bringing a QI project forward and maintaining QI in practice. 

In this study, we examine the role of hospitalists in training within QI projects. To this end, we evaluate how hospitalists in training shape their leadership role within QI projects with a focus on the role of credibility. The hospitalist role is a relatively new speciality that was introduced in the Netherlands in 2012. The Dutch approach to the hospitalist concept has been inspired by the hospitalist model used in the US and Canada, but is unique in that QI has been integrated as a core competence of the hospitalist [14,15,16]. To date, much of the literature on how to lead QI projects and enact the role of champion has focused on physicians who have already attained a leadership position [1,3,17,18]. This study aims to unravel how hospitalists in training shape their role as a champion within the context of QI projects in hospital care, and why some are more effective in leading a QI project than others. We focus on the role of credibility, as this is a prerequisite for successfully filling the role of champion. This aim is translated into the following research questions:How does the credibility of hospitalists in training influence the outcome of their QI projects?What attributes of credibility are important for a hospitalist in training to successfully accomplish a QI project?What supportive contextual factors can contribute to the hospitalist in training’s championing role of successfully leading a QI project?

## 2. Materials and Methods

### 2.1. Design and Approach

A multiple-case design was used as appropriate given the descriptive and explanatory nature of the research aims [19]. The role of the hospitalist in a QI project served as the unit of analysis. To facilitate comprehensive reporting, we have used the Consolidated Criteria for Reporting Qualitative Research checklist (COREQ) [20].

### 2.2. Participants

Each hospitalist in training has to conduct a project in an area of QI within their clinical training setting [21]. These QI projects should contain a clinical audit to evaluate delivered care in terms of the current standard and a re-audit to demonstrate that improvements have been made. We purposively selected two cases from each hospital based on the following inclusion criteria: The QI project was conducted by a hospitalist in training since 2016 to reduce recall bias, andThe QI project had been completed and presented to stakeholders of the hospital ward involved.

All QI projects are supported by a quality officer and supervised by a member of the medical staff. The quality officer has expertise in quality of care and patient safety management and was closely involved in the process and assessment of the QI project. The supervising member of the medical staff was less closely, but sufficiently, involved during the project to be able to give an objective final assessment of the QI project and its outcome. 

### 2.3. Data Collection

Data were collected through individual face-to-face and online semi-structured interviews by two researchers (GW and LH-M) between October 2019 and February 2021. Most interviews were conducted online due to restrictions related to the COVID-19 pandemic. Reports on the QI projects were read before the interviews were conducted to understand the context and the role of the hospitalist in training during the QI project. The interview guide was developed by the authors (RG, KA, GW, LH-M) based on the results of a realist review on audits [1]. The interview guide included questions on the role in the QI project of the hospitalist in training. To improve the accuracy of the recalled data, particularly concerning the time at which events happened, we used a life grid approach during the interview in which events that occurred during the QI projects (e.g., training colleagues about the QI project) were related to significant external events (such as the wedding of Prince Harry and Meghan Markle or the Ebola outbreak in the Democratic Republic of Congo) [22]. During the interviews, we asked respondents to describe the context in which the hospitalist in training worked, how the hospitalist in training carried out the role of champion during the QI project, and to what extent improvements suggested by the QI project were implemented. The supervisor and quality officer were specifically asked about how the hospitalist in training was able to persuade and team up with other involved healthcare professionals and whether stakeholders listened to them and whether the QI project was successful, i.e., whether the QI suggestions were adopted in clinical practice. 

### 2.4. Data Analysis

The interviews were recorded, transcribed, and pseudonymized. Interview transcripts were entered into ATLAS.ti (version 9, Scientific Software Development GmbH, Berlin, Germany) for data management and analysis [23]. The qualitative data analysis was iterative in order to build explanations of events over time [19]. Two researchers (GW and LH-M) analysed six interviews independently to identify thematic categories corresponding with the topics investigated while creating new codes for emergent themes. The originally expected themes were the positioning of the hospitalist in training within the organization, their leadership skills, and their credibility. These themes were reviewed and refined and new themes were added as they emerged from the data. The researchers met to form a consensus on the interpretations of existing codes, to compare coding, to discuss emerging themes and to integrate these into the coding framework. One researcher (LH-M) coded the remaining interviews using this coding framework. 

We first coded data for each case, and then described the different contexts of each case in detail to ease comparison across settings [19,24]. We then compared the cases and contrasted the attributes of credibility. The research team retained reflexivity by discussing and challenging the patterns that emerged.

## 3. Results

In total, 23 participants were interviewed (see Table 1). Seven of the nine (78%) hospitalists in training were female, which is in line with the percentage of female physicians in training in the Netherlands. At the time of the interviews, the mean age of the hospitalists in training was 33.3 (SD = 3.3, ranging from 30–40 years), and the mean number of years since graduation was 7 (SD = 1.7, ranging from 5–10 years).

Our findings are organized into two sections. The first section primarily describes how and why hospitalists in training earned and were perceived credible in the eyes of colleagues, and how and why each attribute of credibility influenced their ability to successfully fulfil the role of champion during their QI project. The second section describes two contextual factors that supported the credibility of the hospitalist in training.

### 3.1. Attributes of Credibility

#### 3.1.1. Being Convincing about the Need for Change by Providing Supportive Clinical Evidence

The first attribute of credibility that emerged from the interviews was being convincing about the need for change by providing supportive clinical evidence. If hospitalists in training were able to get medical specialists on board by showing data that supported the need for the QI project, then their credibility increased among quality officers and supervisors. One quality officer stated the following about a hospitalist in training:

“What she could convince us with were the data she had collected. She was able to rise above the numbers and understood what was going on. She was well prepared. (…) She was credible, mainly because—and especially medical specialists require that—she came up with good data.”(Quality officer03, hospital C)

Some hospitalists in training also mentioned that they sometimes had difficult discussions with medical specialists to convince them of the urgency of the intended change, as medical specialists could be reluctant to change their practices:

“Within some departments, we encountered a lot more resistance, but at that time we could show how many errors had been eliminated by changing care practices. The availability of data made it easier to convince medical specialists.” (Hospitalist06, hospital C)

#### 3.1.2. Displaying Competence in Their Clinical Work and Commitment to Their Tasks

The second attribute of credibility was established by hospitalists in training displaying commitment to their clinical work and being persistent in completing their work. Many hospitalists in training argued that they had to earn credibility with medical specialists during the QI project by building trust. This trust was built by working in a ward for a longer period of time and by demonstrating clinical expertise and that they were someone others could rely on. In this way, hospitalists in training felt that they had earned credibility and felt able to voice their opinion with medical specialists about the subject of the QI project:

“Right about the time I had all my data, which was about the fifth month of my rotation, I was in a position to voice my opinion. In the first month, or couple of months, I was not yet in a position to say anything, because you’ve not been there long enough. If you’re not yet part of a team, then it’s not really accepted that you’re going to tell them something you think they have to change.”(Hospitalist01, hospital A)

“You have to engage in and lead difficult conversations with the medical specialists about the subject of the QI project. (…) When hospitalist04 said something, every medical specialist listened. I noticed that medical specialists had a lot of respect for her because of her competence and intelligence. All of them had loads of experience, but they listen to her incredibly seriously.”(Supervisor03, hospital B)

Hospitalists in training felt that they gained respect and trust from the nursing staff for their role in the coordination of care and by being approachable and available in the ward at all times. Nursing staff valued and trusted the hospitalist for being a stable factor in the ward for patient care. With this trust and respect, hospitalists in training felt that they were able to get nurses on board to sustain a change they had suggested:

“I think I was also able to convince the nurses because they appreciated my role as healthcare professional. Because they saw what I did for the patients, things that the ENT resident did not or could not do, the nurses also thought, “Well, if she can do all that, then the QI project will also be relevant”.”(Hospitalist01, hospital A)

When asked how they built their credibility and were subsequently successful in bringing the QI project to the next stage, many hospitalists in training cited working in collaboration with nurses on the ward as one of the main reasons. In addition, hospitalists in training were aware of the importance of earning credibility and buy-in from nurses since they needed them to advance the QI project once the hospitalist in training had finished their time in the ward.

#### 3.1.3. Generating Shared Ownership of the QI Project with Other Healthcare Professionals

Supervisors and quality officers described those hospitalists in training who were successful in advancing their QI project as having identified and addressed the appropriate group of stakeholders. In addition, quality officers and supervisors viewed such hospitalists in training as people who can inspire and motivate stakeholders so that they share the responsibility for the outcome of the QI project, something that is important to initiate and sustain the changes that emerge from a QI project.

According to the respondents, it was crucial that the whole ward team was involved in the QI project, as the team had to accept the suggestions for change and then sustain them. If the team granted credibility to the hospitalist in training, then the hospitalist in training was not only able to show all stakeholders the importance of the QI project but also to secure and sustain the changes after the QI project was finished by having ensured their commitment. For example, hospitalist06 was able to bring the nursing team on board for this specific QI project, and therefore the changes were successfully maintained as the team took ownership and responsibility for the changes:

“The hospitalist in training was held in high esteem by the nursing team. Further, she took the team along with her to bring about the changes and they were so pleased with them, and that has continued ever since. This is because the team was so intensively immersed in the whole QI project. After the departure of the hospitalist in training, two nursing team leaders clearly took ownership of the QI project … and they also feel a real sense of ownership.”(Quality officer03, hospital C)

#### 3.1.4. Acting as a Team Player to Foster Collaboration during the QI Project

This attribute describes the way credibility was built by hospitalists in training by positioning themselves within the team. Many hospitalists in training who fostered contributions from others in their QI project were described as a "team player" by supervisors and quality officers. For example, hospitalists in training who were successful in advancing their QI project engaged nurses in decision-making about how to embed the changes within existing nursing care practices or routines, thereby emphasising that the opinions of all team members were essential and that all stakeholders were valued as contributing to the QI process: 

“I am a real team player by nature. I think it helped a lot that I got to know the structure of that ward pretty quickly. As a team player, I have got to know people and also know the qualities they have and how to express appreciation for their contributions. In this way, I also knew how to convey the message what was great about the improvement from the perspective of the nurses and how to put this into words. I studied psychology before, so I can use words to motivate people quite well.”(Hospitalist01, hospital A)

On the other hand, some hospitalists in training were not able to motivate other professionals and were not successful in bringing their QI project forward. Quality officers and supervisors described those hospitalists in training as "soloists", meaning that they worked mainly on their own and found it hard to work together with other stakeholders during their QI project:

“What I especially noticed with hospitalist05 is that he worked very much on his own from the start. Whereas hospitalist06 worked as member of a team, he worked much more as a loner. And, because of that, I noticed that his subject was supported much less by the team. (…) He had generated little support for the QI project.”(Quality officer03, hospital C)

The relatively short period of six months that hospitalists in training work on a ward was sometimes considered by the hospitalists in training themselves as an impeding factor for setting QI changes in motion, for working together and, more importantly, for earning credibility and inspiring stakeholders to "buy-in". As a consequence, a QI project might not be well supported by nurses or medical specialists and, because of this, they sensed little ownership of the QI project and did not prioritize the advancing of the QI project in the longer term:

“When you arrive at a ward for the first time, you are not inclined to say, “Can you do that for me?” Because you do not know the people, you are inclined to do everything yourself. (…) But I think that if I had given people a few more tasks, then it would have been more of a shared QI project and perhaps the subject would have gained more attention in the ward, and perhaps it would have continued a little longer after my departure, but that’s all hindsight. I think that this is a disadvantage of the way I approached my QI project. It was my project and I worked very hard on it, but I could not convey the urgency to others…. Everyone felt the urgency, but apparently not urgently enough.”(Hospitalist08, hospital E)

“You are a guest on the ward, and thus also an outsider. And you are also the one who has come to change something, but this can only happen if they [the professionals on the ward] want the changes to happen. You don’t just get a project group or a fellow doctor who wants to change things with you, that can be quite difficult.”(Supervisor04, hospital C)

### 3.2. Contextual Factors Related to the Credibility of the Hospitalist in Training

We also found two contextual factors that supported the credibility of the hospitalist in training: (1) choosing a subject for the QI project that was perceived as urgently required by the group of stakeholders involved, and (2) being supported by the board of directors and other formal and informal leaders as the leader of a QI project.

#### 3.2.1. Choosing a Subject for the QI Project That Was Perceived as Urgently Required by the Group of Stakeholders Involved

Choosing a subject for a QI project that is relevant and perceived as urgent from the perspective of all stakeholders was crucial for the success of a QI project. Some QI projects were considered more urgent by the nursing team (such as dealing with sleeplessness in admitted patients) while other projects were more pressing for medical specialities (e.g., the dosing of vitamin K antagonists). Clearly, although both groups of stakeholders work in the same ward, they can have different senses of urgency for a particular QI project. One hospitalist in training stated that the sense of urgency between medical specialists and the nursing team diverged substantially for his QI project, as the nursing team was more affected by the subject: 

“For example, during the ward round with the medical specialist, I saw a patient who said, “I haven’t slept.” And then, a number of doctors would say, “That’s just how it is in a hospital”. (…) From a nursing perspective, they are the ones that have to deal with the patient night and day. Then nurses are often told by doctors that they accept sleeplessness in patients. (…) So I think that the urgency in solving this problem, ensuring a good night’s rest for the patient, lies mainly with the nurses. The doctor hangs up the phone, so to speak, and has already forgotten the patient. They are not bothered that much by a sleepless patient.”(Hospitalist02, hospital A)

In this example, it was crucial that the hospitalist in training chose a subject for the QI project that was perceived as urgent by the nursing team as this earned her credibility from the nursing team and inspired them to sustain the changes after the QI project finished. Overall, it was crucial for hospitalists in training to gain insight into where the changes related to their QI project needed to take place and to identify the group of stakeholders who experienced a sense of urgency related to the QI subject.

#### 3.2.2. Being Supported by the Board of Directors and Other Formal and Informal Leaders as the Leader of a QI Project

The boards of directors of all the hospitals that had introduced the function of hospitalist were enthusiastic about this new healthcare role with a special focus on continuity of care and QI. This created a supportive context for hospitalists and hospitalists in training, with the board of directors communicating the added value of hospitalists and hospitalists in training for continuity and for QI within the hospital. Support by other formal and informal leaders among the medical specialists was also important. For example, respondents from Hospital D commented that medical specialists who saw the potential added value of a hospitalist were the driving force in mobilizing broad support for the introduction of this health care professional within their hospital. The hospital board of Hospital C had truly embraced the concept of hospitalists, and had more hospitalists in training than the other hospitals participating in this study. Having more hospitalists in training helped this hospital generate awareness of this new function and led to its acceptance among medical specialists, which served as a basis for gaining credibility for this new function. At Hospital C, the respondents indicated that within the board of directors there was a clear vision and sense of urgency for appointing hospitalists, and the training of hospitalists was financed by the hospital itself. Furthermore, one supervisor at this hospital was a prominent driving force for securing a solid place for the hospitalist in patient care:

“I had been a supervisor in the internal medicine training programme for a while. So I reasoned that, with all that experience, it would be nice to help roll out the idea of the hospitalist myself. From the very start, I played a role as a direct supervisor/educator, but at the same time took responsibility for the quality of the training of the hospitalists. (…) We were prepared to finance the training for hospitalists ourselves because we also needed to appoint these hospitalists. And that’s why we trained many more hospitalists in the beginning than the other hospitals.”(Supervisor04, hospital C)

## 4. Discussion

This study set out to deepen our understanding of how hospitalists in training shape their role as a champion within the context of QI projects. A summary of findings is provided in Table 2. We identified four attributes of credibility that were essential for the hospitalist in training to succeed as a champion: (1) being convincing about the need for change by providing supportive clinical evidence, (2) displaying competence in their clinical work and commitment to their tasks, (3) generating shared ownership of the QI project with other healthcare professionals, and (4) acting as a team player to foster collaboration during the QI project. We also identified two contextual factors that supported the credibility of the hospitalist in training: (1) choosing a subject for the QI project that was perceived as urgently required by the group of stakeholders involved, and (2) being supported by the board of directors and other formal and informal leaders as the leader of a QI project. Our findings help clarify how the credibility of the hospitalists in training shapes their role as a champion within QI, providing evidence that these credibility attributes can affect the outcomes of QI projects.

Previous studies have shown that the collaborative skills of physicians leading QI projects and a collegial disposition are very important in gaining credibility [12,25]. Indeed, collaborative skills are often mentioned as necessary for physicians to engage colleagues and managers in effecting changes in hospital care [12,26]. Our study adds depth to these findings by showing that it is not just a collegial disposition but also the ability to engage a multidisciplinary team that are important preconditions for fostering collaboration during a QI project and earning credibility. Hospitalists in training who were successful champions were often described as ‘a team player’ by being able to emphasise the qualities of all members of the team. In addition, they were able to generate support from all stakeholders by ensuring that the opinions of all members of a team were valued and that a consensus was reached about how to embed change in current practice and to sustain changes after the QI project ended. Our findings regarding the importance of multidisciplinary teamwork and collaboration are in line with previous research, showing that establishing trust and valuing the contributions of others are necessary to function effectively as a team and also helps to build collaboration [27]. This is particularly powerful as it brings various perspectives on QI together, generates shared ownership of the QI project and prioritises advancing the QI project in the longer term. 

Our findings that hospitalists in training felt a need to earn credibility and build trust among medical specialists over time by working in a ward for an extended period and by demonstrating clinical expertise and reliability, resonates with the literature on entrustment of decision-making in clinical training by medical specialists [28,29,30]. Hauer et al. (2014) stated that trust “acts as a gatekeeper to the learner’s increasing level of participation and responsibility in the workplace” [30]. As such, it is not surprising that participants in our study mentioned that medical specialists first needed to be convinced about the clinical expertise of the hospitalist in training given that this is related to the medical specialist’s role as supervisor. Supervisors need to make decisions about how much independence over patient care tasks can be granted to a physician in training while ensuring the quality of patient care while, at the same time, giving physicians in training appropriate and progressively greater responsibility. Being entrusted with increasing clinical autonomy has everything to do with the degree of trust that the medical specialist has in the physician in training regarding patient care and it seems this translates to the trust they are granted for their QI improvement project. 

Providing health professionals with feedback data on their clinical performance when treating specific groups of patients has long been used as a QI strategy [1,31,32]. A previous review of the mechanisms of audits concluded that the use of data was important in enabling healthcare professionals to identify shortcomings in their local patient care and strengthen their confidence when discussing requests for changes with “those in positions of leadership” [1]. We saw that data were often intentionally used by hospitalists in training to earn credibility and to get healthcare professionals on board. Hospitalists in training earned credibility by providing sound data that supported their expertise in QI as well as their own role as a champion. 

In this study, we focused on how hospitalists in training fulfil the role of champion during their QI project. We observed that the support of the board of directors during the QI projects was important and influenced the support for the new function among medical specialists as the hospitalist function is relatively new and not always fully accepted by medical specialists. Previous studies have described how, as a young professional, leading a QI project can be challenging for physicians in training, as they are in vulnerable positions within the medical education hierarchy [33,34,35]. Interestingly, the results of our study suggest otherwise: a supportive environment does not seem to be essential for a hospitalist in training to fill the championing role provided the four attributes of credibility were present. However, if the hospitalist in training lacked any of the four attributes of credibility, then a supportive environment would not be sufficient for them to succeed as a champion. This strengthens the claims that credibility is a prerequisite for successfully fulfilling the role of champion. 

### 4.1. Implications 

Our study uncovered a broad range of attributes of credibility that were critical factors for a hospitalist in training in their role as a champion in a QI project. Our findings translate into one theoretical and three practical implications.

The theoretical implication is that this study adds clarity to the concept of credibility of hospitalists in training. The analysis presented here affirms the call by others for a more nuanced conceptualization of the credibility of champions within QI projects and the context in which champions enact [3,4,17,36]. Future studies could examine how residents develop themselves over time as champions and how their credibility as a champion influences the quality and efficiency of care.

The first implication for practice is that our findings can be used to improve medical education and QI programmes. Based on our findings, postgraduate training and training in QI for physicians in training should focus more on personal development, specifically on collaboration skills such as obtaining multidisciplinary stakeholder involvement, developing a clear vision on QI and being able to convey the need for change to other healthcare professionals with a focus on shared learning. These factors would seem to be more important than merely focusing on management skills or imparting knowledge of healthcare systems. Often, it is implied that effective champions have certain intrinsic qualities that cannot be taught [8,17,37]. However, a recent article on the attributes of effective champions suggested that many of the necessary skills can be learned [3]. In line with this, we would argue that many of the necessary attributes identified in this study can be learned, and that supporting the development of these skills may be key to sustaining changes. 

The second practical Implication is that the results of this study could contribute to the promotion of the hospitalist role as an important healthcare professional with expertise in QI. In the process of selecting hospitalists in training, organizations and program directors should focus on selecting physicians with strong interpersonal skills in terms of communication and collaboration within multidisciplinary teams and ones who have a strong vision and the ability to convey this to others. This has the potential to lead to a better fit between the hospitalist in training and their task of leading and accomplishing a QI project.

The third practical implication is that credibility seems to accumulate over time, similar to how trust is built among supervisors deciding to what extent to trust residents to carry out patient care on their own. We found that hospitalists in training were perceived as credible by other healthcare professionals when they act as a ‘team player’, emphasising that the opinions of all team members were essential and that all stakeholders are valued as contributing to the QI process. These results confirm those of other studies about how trust and entrustment of residents accumulates over time in supervisors. Wijnen-Meijer et al., (2013) found that "teamwork and collegiality" was one of the factors considered most important for entrustment decisions by supervisors [38]. Further research is needed to expand on the results presented in this study, for example, by gaining a deeper understanding of supervisors’ experience in assigning responsibilities by entrustment to learners and the influence of the supervisor-resident relationship therein. 

### 4.2. Strengths and Limitations

In terms of strengths, we optimized the quality of our design, analysis and interpretations by adopting an iterative approach in which we critically reflected on the research process as it developed. The numerous discussions, reflections and conversations have resulted in a richer overall outcome. Additionally, our findings are based on the situation found in five different hospitals. This is seen as a strength because we explored the credibility of hospitalists in training in different contexts, which can contribute to the transferability and generalizability of our findings to other settings [19,24]. 

Our study also has its limitations. First, given that this qualitative study was designed and conducted after the investigated QI projects were completed, recall bias may be present. We sought to minimize recall bias by selecting QI projects that were conducted relatively recently (after 2015) and by using a life grid approach during the interviews [22]. Second, the limited sample size does not allow us to determine which attributes of credibility are either necessary or sufficient, nor can we isolate the effects of individual attributes. However, we did achieve data saturation in terms of the four attributes of credibility after 23 interviews. Including participants from both academic and non-academic teaching hospitals that are seen as early adopters and leading the introduction of the hospitalist speciality in the Netherlands further contributed to the representativeness of the study population. Overall, we believe that the contextual variations across the different hospitals and the robust qualitative methodology have countered concerns over the limited sample size and provided a rich and new understanding of how attributes of credibility may help physicians in training fill the championing role within QI.

## 5. Conclusions

This study has identified several attributes of credibility that were decisive factors for a hospitalist in training in their role as a champion in a QI project. Hospitalists in training were able to convince other healthcare professionals of the need for change by providing data as supportive evidence and by displaying commitment to their clinical work. In addition, they were able to advance their QI project by generating shared ownership of the QI project with other healthcare professionals and by acting as a team player to foster collaboration during the QI project. Two contextual factors could support the hospitalist in training in advancing their QI project: choosing a subject for the QI project that was perceived as urgent by the group of stakeholders involved, and gaining the support of the board of directors and other informal leaders for the hospitalist as the leader of the QI project. These findings increase our understanding of how hospitalists in training shape their role as a champion and why some are more effective than others in leading a QI project. We believe that many of the necessary attributes identified in this study, such as communication and collaboration skills, can be learned. Our findings could support healthcare organizations in selecting and preparing healthcare professionals for leading change efforts in healthcare.

Future research could prospectively evaluate potential causal relationships between attributes of credibility and change sustainability. While the focus of this study was on the role of hospitalists in training during their QI project, it might be beneficial to include other healthcare professionals who have a championing role, as this could further elucidate and refine attributes of credibility within professional roles.

## Figures and Tables

**Table 1 ijerph-19-16335-t001:** Overview of the study respondents.

Hospital	Respondents
A	Hospitalist01
	Hospitalist02
	Supervisor01
	Supervisor02
	Quality officer01
B	Hospitalist03
	Hospitalist04
	Supervisor03
	Quality officer02
C	Hospitalist05
	Hospitalist06
	Supervisor04
	Quality officer03
D	Hospitalist07
	Supervisor05
	Quality officer04
E	Hospitalist08
	Hospitalist09
	Supervisor06
	Supervisor07
	Quality officer05
	Quality officer06
	Quality officer07

**Table 2 ijerph-19-16335-t002:** Summary of findings.

Attributes of Credibility
Being convincing about the need for change by providing supportive clinical evidence
Displaying competence in their clinical work and commitment to their tasks
Generating shared ownership of the QI project with other healthcare professionals
Acting as a team player to foster collaboration during the QI project.
**Contextual Factors**
Choosing a subject for the QI project that was perceived as urgently required by the group of stakeholders involved
Being supported by the board of directors and other formal and informal leaders as the leader of a QI project

## Data Availability

The datasets generated and analysed during the current study are not publicly available because study participants did not explicitly agree that their raw data would be shared publicly but are available from the corresponding author upon reasonable request.

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
