# Peer review of "Which Attributes of Credibility Matter for Quality Improvement Projects in Hospital Care—A Multiple Case Study among Hospitalists in Training"

_ijerph, 2022, doi:10.3390/ijerph192316335_

Round 1

Reviewer 1 Report

It will enhance the research novelty if the theoretical implication is included since many readers would like to know how a  selected theory governs the results of the study, particularly related to QI, from a qualitative approach.

To add value to the findings, a summary table of findings (i.e. related to themes development) or figures will benefit new researchers.

Author Response

  1. It will enhance the research novelty if the theoretical implication is included since many readers would like to know how a selected theory governs the results of the study, particularly related to QI, from a qualitative approach.

Thank you for your interest in our paper and for taking the time to express your feedback and suggestions. We have added a theoretical implication to the discussion on page 9 and 10:

“The theoretical implication is that this study adds clarity to the concept of credibility of hospitalists in training The analysis presented here affirm the call by others for a more nuanced conceptualisation of the credibility of champions within QI projects and the con-text in which they enact [36-39]. Future studies could examine how residents develop themselves over time as champion and how their credibility as a champion influence quality and efficiency of care.”

  1. To add value to the findings, a summary table of findings (i.e. related to themes development) or figures will benefit new researchers.

Thank you for raising this point. We have incorporated a table with a summary of findings on page 9.

Reviewer 2 Report

Thank you for the opportunity to read and review this very interesting and well-written study. The gap in the literature is persuasively specified and addressed in a rigorous manner. The results are highly relevant. I believe that this paper has much to offer to the audience and should be published with only small amendments. These amendments are:

1.     Table 1 is a little bit unclear. In hospital A, does it not suggest that three respondents were interviewed? If so, why are we only listing Hospitalist 01 and 02? I also note that some quotes refer to informants not in this table such as “supervisor, hospital C”. Please do more to make this table more readable, and informative.

2.     I believe that more could be said about how you approach the notion of credibility. Credibility seems to relate to what the hospitalists do, but it is also something that exists in the eyes of others (whatever the hospitalist might be doing). There is an important time element: it is something that is sort of accumulated. Perhaps you can elaborate further on how you understand credibility? For instance, can you show a connection between what hospitalists do, and how they are perceived? This would add to the practical implications of the study, and perhaps offer further avenues for future research.

3.     What does ‘established coding strategy’ p. 3, l. 130 refer to?

Author Response

Thank you for the opportunity to read and review this very interesting and well-written study. The gap in the literature is persuasively specified and addressed in a rigorous manner. The results are highly relevant. I believe that this paper has much to offer to the audience and should be published with only small amendments. These amendments are:

  1. Table 1 is a little bit unclear. In hospital A, does it not suggest that three respondents were interviewed? If so, why are we only listing Hospitalist 01 and 02? I also note that some quotes refer to informants not in this table such as “supervisor, hospital C”. Please do more to make this table more readable, and informative.

Thank you for your valuable feedback; it helps us to improve our explanation of and the reader’s understanding of our study. Indeed, Table 1 is a bit unclear. We’ve added further clarification by adding numbers to the supervisors and quality officers in Table 1 as well as in the text.

  1. I believe that more could be said about how you approach the notion of credibility. Credibility seems to relate to what the hospitalists do, but it is also something that exists in the eyes of others (whatever the hospitalist might be doing). There is an important time element: it is something that is sort of accumulated. Perhaps you can elaborate further on how you understand credibility? For instance, can you show a connection between what hospitalists do, and how they are perceived? This would add to the practical implications of the study, and perhaps offer further avenues for future research.

You have raised an important point. We believe that credibility might grow over time, similar to how trust is built among supervisors deciding how far to trust residents to carry out patient care on their own. Our study found, for example, that hospitalists in training were perceived as credible by other healthcare professionals when they act as a ‘team player’, emphasising that the opinions of all team members were essential and that all stakeholders were valued as contributing to the QI process. These results confirm those of other studies about how trust and entrustment accumulates over time by supervisors of residents. For example, Wijnen-Meijer et al., (2013) asked experienced clinical educators in the Netherlands and Germany which general resident features lead them to trust residents to perform critical tasks. ‘Teamwork and collegiality’ was one of the factors considered most important for entrustment decisions by supervisors of residents. We have added more details in the discussion on page 10:

“The third practical implication is that credibility seems to accumulate over time, similar to how trust is built among supervisors deciding to what extent to trust residents to carry out patient care on their own. We found that hospitalists in training were perceived as credible by other healthcare professionals when they act as a ‘team player’, emphasising that the opinions of all team members were essential and that all stakeholders are valued as contributing to the QI process. These results confirm those of other studies about how trust and entrustment of residents accumulates over time in supervisors. Wijnen-Meijer et al., (2013) found that ‘Teamwork and collegiality’ was one of the factors considered most important for entrustment decisions by supervisors [41]. Further research is needed to expand on the results presented in this study, for example, by gaining a deeper understanding of supervisors’ experience in assigning responsibilities by entrustment to learners and the influence of the supervisor-resident relationship therein.”

Additionally, we have added our conceptualisation of credibility on page 2:

“From our conceptualisation, credibility can be built over time and calls for the quality of being believable in the eyes of others, which is important to develop trust.”

  1. What does ‘established coding strategy’ p. 3, l. 130 refer to?

Thank you for addressing this point. The ‘established coding strategy’ refers to the coding framework developed by the two researchers (GW and LH-M). We agree with you that this description is unclear and revised the sentence, which now reads:

“One researcher (LH-M) coded the remaining interviews using this coding framework.”